# Composite Hydrogels with Included Solid-State Nanoparticles Bearing Anticancer Chemotherapeutics

**DOI:** 10.3390/gels9050421

**Published:** 2023-05-17

**Authors:** Alexandar M. Zhivkov, Trifon T. Popov, Svetlana H. Hristova

**Affiliations:** 1Institute of Physical Chemistry, Bulgarian Academy of Sciences, Acad. G. Bonchev Str., bl. 11, 1113 Sofia, Bulgaria; 2Medical Faculty, Medical University—Sofia, Zdrave Str. 2, 1431 Sofia, Bulgaria; 3Department of Medical Physics and Biophysics, Medical Faculty, Medical University—Sofia, Zdrave Str. 2, 1431 Sofia, Bulgaria

**Keywords:** hydrogels, nanoparticles, anticancer chemotherapeutics, cancer cell cultures, cytotoxicity

## Abstract

Hydrogels have many useful physicochemical properties which, in combination with their biocompatibility, suggest their application as a drug delivery system for the local and prorogated release of drugs. However, their drug-absorption capacity is limited because of the gel net’s poor adsorption of hydrophilic molecules and in particular, hydrophobic molecules. The absorption capacity of hydrogels can be increased with the incorporation of nanoparticles due to their huge surface area. In this review, composite hydrogels (physical, covalent and injectable) with included hydrophobic and hydrophilic nanoparticles are considered as suitable for use as carriers of anticancer chemotherapeutics. The main focus is given to the surface properties of the nanoparticles (hydrophilicity/hydrophobicity and surface electric charge) formed from metal and dielectric substances: metals (gold, silver), metal-oxides (iron, aluminum, titanium, zirconium), silicates (quartz) and carbon (graphene). The physicochemical properties of the nanoparticles are emphasized in order to assist researchers in choosing appropriate nanoparticles for the adsorption of drugs with hydrophilic and hydrophobic organic molecules.

## 1. Introduction

Cancer is a socially significant disease that is the second leading cause of death worldwide after cardiovascular diseases. For instance, in 2020, almost 10 million people died of oncological disease [1]. In developed countries, the incidence rate of cancer is steadily increasing (around 3–5% per year), this is a silent pandemic [1,2]. Conventionally, chemotherapy and/or radiotherapy are used in clinical practice for the treatment of neoplasms, but some cancer types are resistant to them and because of this, it is necessary to search for new treatment approaches [3,4]. The main problem is that the chemotherapeutics attack all dividing cells, including those of the immune system, which lead to immunosuppression and as a result, there is risk for the patient to die from a banal bacterial or viral infection [5,6]. This problem can be mitigated if chemotherapeutics are administered locally so that their concentration is high in cancer tissue and low in healthy tissue [7,8]. Another problem is that the intracellular concentration of chemotherapeutics fluctuates depending on the frequency of administration (injectable or oral), and this requires the administration of higher concentrations, at which the toxic effect is particularly strong. To avoid this problem, a chemotherapeutic depot should be used so that its concentration over time is maintained at an optimal therapeutic level [9]. For this purpose, different types of chemotherapeutic carriers, such as nanoparticles [10,11,12], hydrogels [13,14], composed nanoparticle–hydrogels [15], micelles [16] and liposomes [17], can be used.

Some problems concerning anticancer therapy can be solved by using hydrogel as a local depot for chemotherapeutics, in particular in the treatment of neoplasms with a superficial localization such as skin cancer, for oral administration in gastric or colon cancer, or of intra-body tumors by injectable hydrogels [18,19,20]. Local administration of effective but highly toxic drugs, such as platinum derivatives, can provide concentrations high enough for effective treatment and minimize their toxic effect on healthy tissues [21]. In addition, the use of gel-depot ensures a uniform penetration of chemotherapeutic agents into the cancer tissue over time, avoiding particularly toxic peak concentrations [22].

To perform its role as a depot, the hydrogel must be able to retain the chemotherapeutic agent and release it gradually as its concentration in the cancer tissue decreases [23]. Chemotherapeutic molecules can be incorporated into the free state, linked by covalent bonds to a biodegradable gel network, or adsorbed onto nano- or micrometer-sized particles (composite gels) [24]. One major advantage of hydrogels is that no convection (caused by different temperatures or concentration gradients) occurs in them, but only the diffusion of molecules is possible, which is, however, slowed down due to the high viscosity caused by the structuring of water molecules into associates around the filaments of the gel network (which are much larger than dynamic nanoassociates in pure water) [25].

A major disadvantage of using hydrogels as a depot for anticancer chemotherapeutics is that most of them are poorly soluble in water [7]. For this reason, incorporation of a chemotherapeutic in the molecular state is inefficient, but can be rectified by mixing a suspension of crystals with the aqueous solution of the polymer (in sol-state before gelation). Another approach to incorporate hydrophobic chemotherapeutics is to use non-homogeneous gels with a hydrophilic network containing hydrophobic domains that arise under certain conditions or are hydrophobic segments of the block copolymer chain [26]. Even for relatively well water-soluble chemotherapeutics, hydrogels do not have significant adsorption capacity because the gel network is composed of linear macromolecules that lack a significant surface area (unlike nanoparticles) necessary for physical adsorption via noncovalent interactions [27]. One possible solution is gels with pH-dependent polymer unit charges that can electrostatically bind oppositely charged chemotherapeutic molecules and release them upon pH change, but this approach faces the following difficulty, that most often, both components have no ionizable groups with pK_a_ in the physiological pH-range and in addition, the pH changes only minimally (in cancer tissue, the pH is lowered by lactic acid accumulation as cells are energized by glycolysis rather than by oxidative phosphorylation). A specific case is that of the covalent binding of a chemotherapeutic, but this is only applicable if the gel network is degraded, for example by enzymes after injection into cancer tissues [28].

Another approach is the incorporation into the gel of nano- or micrometer-sized particles with adsorbed chemotherapeutic molecules that are gradually released spontaneously (according to the equilibrium constant and their local concentration), or induced (e.g., by thermal or photothermal action). The high surface area/mass ratio of the nanoparticles provides a sufficiently large physical adsorption, and their electrical charge (intrinsic or added by the chemical modification of their surface) causes pH-dependent electrostatic adsorption during gel preparation and desorption into the cancer tissue. For adsorption of hydrophobic chemotherapeutics, the particle surface can be chemically modified to become hydrophobic, and after adsorption, coated with a surfactant (low molecular weight or polymeric) so that the composite particles become water-soluble for incorporation into the hydrophilic gel network [28].

Recent studies have demonstrated that nanoparticles interact with biological media (blood plasma, intra- and extracellular liquid) by forming a protein corona on their surface [29,30]. This alters the physicochemical properties of nanoparticles and should be considered when nanoparticles carrying chemotherapeutics are applied in cancer therapy [31]. The effect of the protein corona depends strongly on the type of plasma proteins (especially the proteins with the highest concentration, such as albumin, globulins—mainly IgG, fibrinogen, apolipoproteins, transferrin, complement factors, etc.) that cover the nanoparticles when they enter the blood after resorption upon systemic administration: enteral (peroral, sublingual, rectal) or parenteral (intravenous, intramuscular, subcutaneous, etc.). The structure and amount of plasma proteins can vary depending on the patient’s comorbidities (as many diseases and conditions can lead to changes in plasma proteomics), family background (genetic factors), lifestyle, geographical factors, etc. It has been proven that people with different diseases and medical conditions (such as diabetes, hypercholesterolemia, rheumatism, hemophilia A and B, thalassemia, breast cancer, hemodialysis, pregnancy, smoking, etc.) form different “personalized protein coronas” [32,33,34]. Consistent with the concept of personalized medicine, the “personalized protein corona effect” must be taken into account when chemotherapeutic nanoparticles are applied in human medicine.

This review is focused on composite hydrogels containing particulate carriers of anticancer chemotherapeutics, which are differentiated according to: (a) the type of gel—physical (reversible heat-induced gelation) and chemical (irreversible gelation by covalent intermolecular bonds); (b) the type of particles (metallic, metal-oxides, (alumo)silicates, graphene oxide); (c) the mode of binding the chemotherapeutic molecules to the particles (physical adsorption by electrostatic or van-der-Waals forces, or chemical by covalent bonds); (d) the mode of release into the cancer tissue (spontaneous physical desorption or pH-induced, thermal, photothermal, or enzymatic or pH-induced degradation of the carrier particles). The review is not intended to cover the entire literature in this area, but to focus on physicochemical mechanisms that are typically under-recognized by the authors of numerous published papers, and in this sense it is a complement to the broad-based review addressing the analytical techniques for quantitative drug detection, which are employed to characterize and evaluate drug release from hydrogels [35].

## 2. Anticancer Agents Used in Chemotherapy

Based on their mechanism of action, chemotherapeutics can be divided into the following groups [36] (Figure 1 and Table 1).

### 2.1. Antimetabolites

Antimetabolites are inhibitors of important enzymes that take part in nucleic acids synthesis. Therefore, cell division is blocked.

#### 2.1.1. Inhibitors of Pyrimidine Synthesis

These antimetabolites are structural analogues of thymine (a pyrimidine nitrogen base included in the DNA) that inhibit the enzyme thymidylate synthase, which leads to a decrease in the concentration of thymidine nucleotides in the cell nucleus. Examples of pyrimidine antimetabolites are 5-fluorouracil, capecitabine and others [47,48]. 5-fluorouracil is slightly soluble in water (solubility = 12.2 mg/mL) and has a weakly basic pK_a_ (pK_a_ = 8.00) [49].

#### 2.1.2. Inhibitors of Purine Synthesis

A major example is 6-mercaptopurine, which is one of the first antitumor agents used for chemotherapy, and in particular for the treatment of acute lymphocytic leukemia (white blood cells cancer). 6-mercaptopurine is a structural analogue of hypoxanthine and is a substrate for the enzyme hypoxanthine/guanine phosphoribosyl transferase, resulting in the formation of 6-thioinosinic acid (TIMP), which blocks the purine biosynthesis in the cell nucleus [61,62]. 6-mercaptopurine is insoluble in water and has a weakly basic pK_a_ (pK_a1_ = 7.77) [63].

#### 2.1.3. Folic Acid Antimetabolites

They are structurally analogous to folic acid (vitamin B_9_, which is important for DNA synthesis and, respectively, for cell division). These antimetabolites inhibit the enzyme dihydrofolate reductase, which transforms folate into an active coenzyme—tetrahydrofolate [37]. One such antimetabolite is methotrexate for instance, which is slightly soluble in water (solubility = 1 mg/mL) and has an acidic pK_a_ (pK_a_ = 4.70) [38,39].

#### 2.1.4. Antimetabolites of Urea

The main representative is hydroxyurea, which blocks the enzyme ribonucleotide reductase (a key enzyme for DNA replication, which converts ribonucleotides to desoxyribonucleotides) [66].

### 2.2. Chemotherapeutics, Directly Modifying the DNA Structure

#### 2.2.1. Alkylating Agents

These chemotherapeutics form reactive alkyl radicals that bind covalently the nitrogen at position 7 in guanine (a purine nitrogen base included in the DNA) within and between adjacent DNA chains, resulting in the disruption of DNA replication and transcription. One such alkylating agent is *cyclophosphamide* [64]. Cyclophosphamide is slightly soluble in water (solubility = 40 mg/mL) [65].

#### 2.2.2. Platinum Compounds

Such chemotherapeutics are cisplatin, carboplatin and others. They bind covalently to the nitrogen at position 7 in guanine, leading to DNA damage in the tumor cells and blockage of cell division [42]. Cisplatin is slightly soluble in water (solubility = 1 mg/mL) and has an acidic pK_a_ (pK_a1_ = 5.37) [43].

### 2.3. Microtubule Inhibitors

Microtubules are non-membranous cell organelles that are part of the cytoskeleton along with microfilaments and intermediate filaments. Microtubules consist of the protein tubulin [67]. Microtubules play a key role in the cell division, as the form the mitotic spindle [68].

#### 2.3.1. Inhibitors of Microtubule Polymerization

These chemotherapeutics bind to the protein tubulin, inhibiting the formation of microtubules (microtubule polymerization). This results in the blockage of mitosis, as the proper chromosomal segregation is impossible without a functional mitotic spindle. Microtubule polymerization inhibitors are vinca alkaloids such as vinblastine, vincristine and others [44,45]. Vinblastine is insoluble in water and has a weakly acidic pK_a_ (pK_a1_ = 5.4) [46].

#### 2.3.2. Inhibitors of Microtubule Depolymerization

These anticancer agents block the microtubule depolymerization, which breaks the equilibrium between polymerization and depolymerization. The excessive stability of microtubules makes them nonfunctional and the spindle apparatus cannot form properly. Examples of microtubule depolymerization inhibitors are taxanes such as *paclitaxel*, *docetaxel* and others [55,56]. Paclitaxel is practically insoluble in water (solubility < 0.1 μg/mL) and has a basic pK_a_ (pK_a_ = 10.00) [57,58], while docetaxel is also insoluble in water and has a basic pK_a_ (pK_a_ = 10.70) [59,60].

### 2.4. Antibiotics

#### 2.4.1. Anthracyclines (Topoisomerase II Inhibitors)

Anthracyclines are inhibitors of the enzyme topoisomerase II, which plays an important role in the DNA replication dealing with the tangles and supercoils of a DNA helix. Such anthracyclines are doxorubicin, epirubicin and others [52,53]. Doxorubicin is soluble in water (solubility = 50 mg/mL) and has a basic pK_a_ (pK_a_ = 8.93) [54].

#### 2.4.2. Bleomycin

This chemotherapeutic, during its metabolism in the body, forms ROS (superoxide and hydroxyl radicals), which attack the 3′-5′-phosphodiester bonds between the nucleotides in the DNA, resulting in DNA strand breakage [50]. Bleomycin is slightly soluble in water (solubility = 20 mg/mL) and has a weakly basic pK_a_ (pK_a1_ = 7.30) [51].

### 2.5. Topoisomerase I Inhibitors

One such inhibitor is camptothecin that blocks the enzyme topoisomerase I, which is essential for DNA replication. This leads to damage of the DNA structure and finally to apoptosis (programmed cell death) of the cell [40]. Camptothecin is practically insoluble in water (solubility = 2.5 μg/mL) and has an acidic pK_a_ (pK_a_ = 4.7) [41].

## 3. Types of Hydrogels

A hydrogel can be defined as a quasi-solid body composed of a three-dimensional (3D) network of hydrophilic macromolecules and water. The weight content of the polymer is drastically less (down to two-three orders of magnitude) than that of the water included in the gel, but should be sufficient to form intermolecular bonds (cross-linking), depending on which types of gels can be divided into physical gels with non-covalent (hydrogen, electrostatic and van-der-Waals forces: London dispersion, permanent and charge-induced dipoles) bonds, and chemicals with covalent intermolecular cross-linking. Convection is absent in hydrogels because the water molecules are structured around the strands of the gel network; only diffusion is possible, allowing them to be used for the delayed release of incorporated drug substances. Hydrogels made from natural polymers of animal and plant origin, such as collagen (a linear protein with a triple polypeptide helix) or amylum (a carbohydrate with a highly branched chain), have been known for centuries, while chemically synthesized ones were developed after 1960 [69]. The advantages of hydrogels as a depot for local application of medicinal substances are due to their tunable physicochemical properties, biocompatibility and the possibility for controllable degradation; due to that, they are intensively investigated as local drug delivery systems [70,71]. Since anticancer chemotherapeutics are poorly or practically insoluble in water (Section 2), hydrogels forming hydrophobic domains that serve as a sink for various hydrophobic drugs are of particular importance [72].

### 3.1. Thermo-Reversible Physical Hydrogels

Such are the classical gels of natural and synthetic hydrophilic polymers, which, after heating and cooling, form a hydrogel that can be repeatedly destroyed and re-formed by thermal action, respectively, when raised above and lowered below the critical gelling temperature. The initial heating is necessary to break the intra-molecular bonds defining the native conformation of the macromolecule, e.g., the triple helix of collagen, and transition the polymer chain to a random coil state, which creates the conditions for the formation of random bonds with adjacent chains forming the gel network upon cooling.

The phase state of the polymer solution is determined by the concentration of the polymer and the properties of the solvent (temperature, pH, ionic composition, water-soluble low molecular weight organic compounds such as alcohols, etc.) [73]. The interactions (intra- or intermolecular) between the polymer units are determined by the thermodynamic properties of the polymer and the solvent (mainly water molecules in the case of hydrogels) expressed by Gibbs free energy ΔG = ΔH − TΔS (enthalpy ΔH and entropy ΔS at temperature T), with the three components of ΔG (polymer–polymer, polymer–water and water–water) determining the temperature range of the hydrogel existence. The balance of forces depends on the ability to form hydrogen bonds (electrostatic H-atom sharing between two electronegative atoms such as O and N, O…H-O in particular) and the presence of whole (Coulomb) and partial (polarized covalent bonds) electric charges, that associate water molecules and determine negative enthalpy ΔH values for hydrophilic groups, or the inability for such bonds; in the last case, only van-der-Waals forces are operative and the orientational order of water molecules around the hydrophobic groups increases (lowering the entropy term ΔS).

The chemical nature of the polymer and the composition of the solvent define a temperature range limited by low and high critical solution temperatures (LCSTs and HCSTs), beyond which the chains are in a random coil conformation (then, the polymer chain occupies 1–3% of the volume of the globule) [74]. As the solvent quality deteriorates, in particular by temperature variation, the size of the polymer globule (defined by the averaged values of the radius of gyration *R*_g_ = 〈*R*_g_^2^〉^1/2^ or the distance *h* = 〈*h*^2^〉^1/2^ between the ends of a linear chain) decreases due to the dominance of polymer–polymer over polymer–water interactions, and the polymer chains may even collapse into a globule. When the polymer concentration is high enough, intermolecular bonds are formed in addition to intramolecular bonds, leading to the formation of a gel network in the temperature range Δ*T* = HCST − LCST; the gel breaks down to a polymer solution when the temperature exceeds the upper threshold (Δ*T* > HCST). Polymer network formation in the gel-forming Δ*T* temperature range is due to two factors acting in sequence: (a) interweaving of the polymer chains in an unfolded conformation (random coil at *T* > HCST or *T* < LCST) due to the high concentration of the polymer solution (the average distance between adjacent chains is less than their gyration radius) and (b) shrinkage of the chains due to solvent deterioration as the temperature increases (at HCST > *T* > LCST), when polymer–polymer interactions predominate.

The balance of forces for some polymers in aqueous solution defines a state of sol at room temperature or below (*T* < 25 °C), a region of gel existence at physiological temperatures (*T* ≈ 37 °C) and reversibly breaking down to a polymer solution when heated above. This offers the advantageous option of injecting the polymer solution into the cancer tissue where the gel forms; such thermosensitive gels are termed injectable [75,76,77,78]; an example of such a polymer is poly(N-isopropylacrylamide) [79]. The gel–sol transition at *T* > HCST allows the gel to be destroyed if necessary, for example by photothermal irradiation with infrared light.

A second type of thermo-reversible physical hydrogels is presented by the chemically synthesized block copolymers with a chain composed of segments with different affinities to water molecules. The choice of copolymers of a different chemical nature in the synthesis provides additional possibilities to achieve suitable physicochemical properties in aqueous solution. For the injectable gels, block copolymers have undoubted advantages over homogeneous (linear or branched) chain polymers, since the choice of the length of the hydrophobic segments allows for achieving a suitable gelation temperature (solution at room temperature and gel at physiological temperature), and the length, flexibility and charge of the hydrophilic segments determine the pH-dependent gelation and an acceptably high in sol-state viscosity, which is an additional problem in injection.

The most common copolymers are linear chain copolymers of the ABA or BAB type, where A and B are hydrophilic and hydrophobic polymers, respectively. In these, the formation of a physical gel is governed by the same factors as in the homogeneous polymers discussed above: high polymer concentration and chain shrinkage as the solvent quality deteriorates, in particular by temperature change. Since hydrophobicity is determined by the total area of atomic groups unable to form hydrogen bonds, increasing the length of the polymer chain segments made of hydrophobic groups, such as methylene groups (-CH_2_-), allows for reaching the gel-state at 37 °C and sol-state at *T* ≤ 20 °C.

The most commonly used hydrophilic polymer is oxipolyethylene (-CH_2_-CH_2_-O-)_n_), which, depending on the molecular mass *M*, is referred to as polyethylene glycol (PEG, low molecular weight, *M* ≤ 3000 g/mol), synthesized without catalyst, electrically neutral) or polyethylene oxide (PEO, high molecular weight, synthesized with a complex metal-organic catalyst and therefore containing bound Ca^2+^ or Zn^2+^ ions that impart a weak positive charge to the chain [80]). The presence of an oxygen atom in the chain backbone determines the high flexibility of the PEG/PEO chain due to increased conformational freedom around the C–O–C bonds, and hydrophilicity is determined by the strong hydration of the oxygen atoms in the ethylene oxide units [81]. The most commonly used block copolymers are ABA and BAB of (A) hydrophilic PEG and (B) hydrophobic polypropylene glycol (PPG, (-CH_2_-CH(CH_3_)-O-)_n_), and its additional methylene group (-CH_3_) determines the hydrophobicity of the PPG segments. Some hydrophobic chain polymers used for the synthesis of block copolymers are shown on Figure 2.

For medical practice, it is important that the sol–gel transition (inducted by the jump from room temperature to 37 °C) does not take place in the needle while the polymer solution is injected by syringe, this can be avoided by using pH as a second factor required for gel network formation (sol-state at pH ≤ 5 and gel-state at pH 7.4) [19,82,83,84]. For this purpose, block-copolymers with hydrophobic segments containing chargeable groups with a constant pK_a_ allows to alter the degree of ionization at transfer to pH 7.4, and by that, the hydrophobicity are used. pH-dependence can be achieved by introducing charged groups into the polymer units with an appropriate dissociation constant (then the hydrophilic segments behave as a polyelectrolyte). 

Polymers with carboxyl groups (pK_a_ ≈ 4) are suitable for this purpose because the pH-dependent dissociation (COOH ↔ COO^−^) emerges in the pH range 3–5, so that at a low pH, the polymer chain is electrically neutral and at pH 7.4, it is negatively charged. The presence of closely disposed ionizable groups (each unit of the chain can carry one or more COO^−^ groups that leads to a high liner charge density) results in a shift of pK_a_ towards the alkaline region due to the increased local concentration of H_3_O^+^ cations and to an anticooperative effect: the curve of the degree of ionization as a function of pH becomes flatter. An example of such a polymer is carboxymethyl cellulose (CMC), which is produced by the chemical modification of natural cellulose (poly-1,4-D-glucose by its chemical nature) at a degree of substitution DS = 0.8–1.2 of the hydroxylic groups of the glucose units with methyl-carboxyl groups (–CH_2_COOH). At degree of substitution DS ≈ 1 (one charge per glucose unit) and pH 7, the high density of negatively charged groups COO^−^ leads to condensation of counterions from the medium (for instance Na^+^ cations), which reduce the effective charge of the chain. A feature of CMC is the high chain rigidity, which is caused by the highly constrained conformational freedom between adjacent glucose units due to the β-1,4-linkage between them (the rotation round C–O–C bonds between the C-1 and C-4 atoms is impossible). This is in contrast to the natural polymer amylose (chemically identical to cellulose), with a flexible chain due to α-configuration at C1 atom in the α-1,4-linkage that allows for rotation between the glucose units.

Other chargeable polymers are those congaing groups of tertiary amines that can obtain a pH-dependent positive charge: ≡N ↔ ≡NH^+^. Such polymers are convenient for injectable gels because the ionization constant of ≡N groups allows for altering their charge in the physiological pH range. An example of such thermo- and pH-dependent injectable gels is the triblock copolymer PAAm-PEG-PAAm [85]. At 20 °C and pH 6.8, the polymer is in a solution-state, but becomes a gel at the physiological 37 °C and pH 7.4 (viscosity increases with more than five orders of magnitude (form 0.1 to 10^4.7^ Pa·s) at a polymer concentration of 12.5 wt%. The pH is the most important factor for the sol–gel phase transition: at 25 °C, the relatively small decrease of the H^+^-concentration from pH 6.8 to pH 7.4 leads to a drastic increase in the viscosity with four orders of magnitude, while the temperature increases from 25 to 37 °C—with only a half order. These parameters are achieved using the appropriate polymer concentration and amphiphilic structure of the triblock copolymer: two long-chain hydrophobic PAAm (poly(amidoamine) segments with dual (pH and temperature) functionality and a medial hydrophilic PEG segments. The pH and temperature increase caused the growth of the hydrophobicity of PAAm segments because of the pH-determined partial deionization of the ≡NH^+^ groups (half of them are charged at pH 7.4 and 20 °C), and additionally by a thermos-induced shift of their ionization constant from pK_a_ 7.4 to pK_a_ 6.8 at temperature increasing from 20 to 40 °C. The increased hydrophobicity of PAAm segments leads to the formation of a gel network by van-der-Waals contacts between the approaching chain parts of the next macromolecules, which are entangled with the segments of adjacent polymer chains (due to the high concentration of the polymer solution).

In the review [13] the sol–gel phase transition of PAAm-PEG-PAAm copolymer (described in Refs. [19,85]) is explained by the transition of PAAc segments from a hydrophilic to hydrophobic state at an increase in pH from low to high values (pH 3.0 → pH 7.4). However, this explanation is incorrect because PAAm is wrongly written as PAAc (poly(acrylic acid) instead (poly(amidoamine); the confusion comes from the fact that in the literature, as least three polymers with quite different structures are designed by the abbreviation PAA (Figure 2). Furthermore, the assertion that a hydrophilic polymer segment with acid groups, such as (poly(acrylic acid), became hydrophobic at ionization (COOH ↔ COO^−^), is principally erroneous, since upon dissociation of a proton from the carboxyl groups they have become even more hydrophilic as the partial charge of the oxygen atom (due to the high electron-affinity, leading to the polarization of the C–O and C–H covalent bonds) becomes a whole (Coulomb) charge, and this leads to an even stronger orientation of the dipole H_2_O molecules, the closest of which are strongly electrostatically bonded to this oxygen atom and form hydrogen bonds with the second layer water molecules, i.e., when the acidity of the polymer solution is reduced from pH 3 to pH 7, the hydrophilic segments become even more hydrophilic, but not hydrophobic.

### 3.2. Irreversible Chemical Hydrogels

This type of hydrogel is produced by the covalent cross-linking of chains of one type of hydrophilic polymer or with chains of another type (copolymers) of hydrophilic or hydrophobic polymer, resulting in a thermo-irreversible gel network. The pore size depends on the length of the polymer chains and the polymer concentration. An example of a small pore hydrophilic gel is poly(acrylamide) (PAM) (used in biochemical studies for the electrophoresis of proteins denatured with the surfactant sodium dodecyl sulfate), with small pores that do not allow for the translational movement of globular proteins in a native conformation.

When the gel network is a co-polymer of hydrophilic and hydrophobic filaments (amphiphilic gel), if the latter have sufficient length, they adopt a collapsed ball conformation—domains—in which hydrophobic chemotherapeutics can be embedded as free molecules or adsorbed onto nanoparticles with a hydrophobic surface [86]. An example of one such gel is a co-polymer of hydrophilic poly(acrylamide) (PAM) and hydrophobic poly(methacrylate) (PMC) [87].

There are several polymerization techniques: free-radical, esterification and photo-polymerization. Free-radical polymerization is only acceptable for externally applied gels, as this type of polymerization always leaves a residual free monomer radical that is highly toxic to tissues. Photopolymerization is used as a second stage of the polymerization of a hydrogel previously formed by free-radical polymerization, so a co-network between poly(methacrylate) (MA) chains crosslinked by ethylene glycol dimethacrylate (EGDMA) is obtained [88].

In Table 2, the most commonly used polymers for hydrogels are given.

### 3.3. Biodegradability of Hydrogels

The biodegradability of the hydrogels has an essential role in the increasing use of hydrogels as a drug delivery system. In the term “biodegradability”, all types of in vivo degradation are included: from simple hydrolysis to enzymatically catalyzed degradation. In the biodegradation process, the polymers that form the hydrogels are degraded to monomers by breaking chemical bonds. There are three main mechanisms of biodegradation [89]: (a) *Solubilization*. A great number of water-soluble polymers are determined as biodegradable due to their ability to dissolve in water. Such polymers are dextran (DEX), poly(ethylene glycol) (PEG), poly(ethylene oxide) (PEO), polyvinyl alcohol (PVA), etc. [90,91]; (b) *Hydrolysis*. Another mechanism of biodegradation of the polymers is the hydrolysis of ester bonds between the monomers with the formation of an alcohol and a carboxylic acid, as well as amide bonds between the monomers with the formation of an amide and a carboxylic acid. Chemical hydrolysis is characteristic for polylactic-co-glycolic acid (PLGA), poly(amidoamine) (PAA) poly(ethylene glycol)-oleic acid (OA-PEG), poly(β-amino ester) (PBAE), etc. [92]; (c) *Enzymatic degradation*. In the human body, there are special enzymes—hydrolases (class III enzymes), which catalyze hydrolytic bonds (carbon–oxygen (C–O), carbon–nitrogen (C–N), carbon–carbon (C–C), phosphorus–nitrogen (P–N) bonds, etc.) and cleavage involving water. For example, hyaluronic acid (HA) is degraded by the hydrolase hyaluronidase [93]. Other polymers that undergo enzymatic hydrolysis and/or pH-sensitive hydrolysis/gelation are chitosan, gelatin, etc. [94,95].

Some polymers used for the composition of hydrogels cannot undergo degradation. Such *non-biodegradable polymers* are cellulose derivatives (for instance: carboxymethyl cellulose, CMC), poly(methacrylate) (PMA), etc. [96].

## 4. Nanoparticles as Carriers of Anticancer Chemotherapeutics

As carriers of anticancer chemotherapeutics, particles of nano- and micrometer sizes and made of different materials can be used: metals, metal-oxides, (alumo)silicates, carbon (graphene and graphene oxide), carbonates, polymeric, denatured proteins, liposomes, etc., or composed of different substances [97,98]; the subjects of this review are the first five types, i.e., solid-state particles. The high dispersity of nanoparticles (determined by their small size) provides a huge specific surface area, reaching hundreds of square meters per gram, and consequently a huge adsorption capacity at the particle/medium interface. In this review, particles are classified according to their constituent substance, since their structure and properties (hydrophilicity/hydrophobicity, surface electric charge, bulk electric polarizability, smoothness/roughness/porosity at the molecular level, etc.) determine their specific adsorption capability (capacity per one unit surface area) for anticancer chemotherapeutics.

The adsorption onto the nanoparticles in aqueous medium emerges because of the difference in the chemical potentials of the two phases (solid and liquid) and leads to the formation of a boundary layer in which the concentration of the adsorbate (small ions, uncharged small molecules, big organic molecules, etc.) is altered compared to the bulk; the dissolved molecules are in competition with water molecules (the main component of the solution) on the particle surface. The adsorption per unit area depends on the physicochemical properties of the solid/liquid interface, the main ones are the hydrophilicity (the affinity to the water molecules) and the surface electric charge. Hydrophilics are the surfaces that are able to form hydrogen bonds (-O…H–OH) with the water molecules such as metal oxides/silicates/carbonates (Section 4.2).

The surface electric charge appears because of the adsorption of small ions from the medium or dissociation of the ionizable groups of nanoparticles [99]. Depending on the solid substance, the potential-determining ions can be metal cations M^+^ (in case of metal in a solution of its salt) or OH^−^ anions and protons H^+^ (existing as hydroxonium cations H_3_O^+^) in case of oxides; the electrolytes that do not charge the surface are called independent. The electrically charged surface attracts the counterions (ions with the opposite sign) and repels the co-ions of both charge-determining and indifferent electrolytes; as a result, an electrical double layer (EDL) is formed on the solid/water interface as a capacitor with another plate consisting of two parts: dense (the counterions are adsorbed on the solid surface) and diffuse (the rest of counterions are dispersed in the vicinity of the surface because of thermal motion). The difference in the ion concentrations (increased for the counterions and decreased for the co-ions, compared to the bulk), and consequently the electrostatic potential in the diffuse layer of the EDL, decrease exponentially (quasi-exponentially at a surface potential above 25 mV) with the distance from the solid surface; the thickness of the EDL is defined as the distance upon which the local electric potential is *e*-times (≈2.72) smaller than that on the solid surface. 

Two main methods are commonly used to calculate the surface charge density: (a) potentiometric titration (measuring the pH at the addition of the acid or base), and (b) free electrophoresis that gives the electrokinetic potential ζ [mV] calculated from the measured electrophoretic mobility μ = (εε_0_/η)ζ = *v*/*E* (the velocity *v* [μm/s] of the particles in the direct electric field with strength *E* [V/cm]) in liquid with relative dielectric permittivity ε (ε_0_ is the electric constant of SI) and viscosity η for water ε = 80 and η = 1 mPa·s at 20 °C). These two techniques give quite different values for the charge density because in the pH-titration, all charges are accessible to the potential-determining ions, but in electrophoresis, the ζ-potential reflects only the charges above the shear plane (which distinguishes between the hydrodynamically mobile and immobile water molecules in the boundary layer); by that, the counterions that are moved together with the nanoparticle in the electric field remain electrophoretically ‘invisible’, such are ions specifically adsorbed by chemical bonds to the surface molecules of the solid phase and the counterions in the dense part of the EDL (electrostatically adsorbed on the surface). As a result, the potentiometrically obtained surface charge density is higher than the electrophoretic one, even in the case of particles with a molecularly smooth surface because the counterions in the dense layer of the EDL are moved together with the nanoparticle (the counterions in the diffuse layer are moved in the opposite direction). The difference between the potentiometric and electrophoretic charge density is large in the case of oxides, owing to the presence of a gelatinous surface layer that has chargeable groups and is permeable to ions [100,101].

The dispersity of the nanoparticles, as their main property which determines their specific surface area (m^2^/g) and their regular distribution in the hydrogel, is limited by the tendency to aggregate in aqueous solutions and fall in sediment (coagulation). According to the aggregational stability (preservation of the nanoparticles’ individuality) the dispersed systems are distinguished into two classes: lyophilic (formed by spontaneously dispersing the substance in the liquid medium) or lyophobic (using energy to split the solid-state body forming an additional surface or chemical reactions to synthesize the nanoparticles). The lyophilic dispersion systems are thermodynamically stable (Δ*G* = Δ*H* − *T*Δ*S* < 0) due to the small difference in the surface energies of the two phases (the particles and the liquid); such are the aqueous suspensions of clays (alumosilicate hydrophilic plates), surfactants (amphiphilic molecules with hydrophilic and hydrophobic parts), polymers with hydroxylic groups such as the cellulose (polyglucose) and other polysaccharides, proteins in native conformation, etc.

The lyophobic dispersion systems are thermodynamically instable (Δ*G* > 0) because of the large difference in the surface energies of the two phases (the solid particles and the liquid medium), but they can be kinetically stable when the interparticle repulsion is stronger than the attraction (dispersion stability); then, the diffusion dominates over the sedimentation (caused by the gravitation) due to the small size of the nano- and colloid particles. A characteristic property of the lyophobic dispersion systems is that the particles aggregate when an electrolyte is added; the aggregation emerges when the ionic strength (determined by the concentration and squared valency of the ions) reaches some critical value; then, the EDL thickness decreases and the electrostatic repulsion between the colliding particles becomes weaker than the van-der-Waals attraction. The repulsion occurs when the ionic atmospheres (diffuse parts of EDL) of neighbor particles are intersected; other stabilizing factors are adsorbed polymers (used for the stabilization of metal nanoparticles) and the gel-like layer on the surface of oxides in aqueous medium. 

The sols (colloidal solutions) and suspensions of solid-state nano- and colloid particles, with both hydrophobic and hydrophilic surfaces, are lyophobic as a rule, i.e., aggregation emerges with increasing electrolyte concentration (potential-determining or indifferent ions) and/or adsorption of molecules with opposite electric charge. The oxides occupy an intermediate position between the lyophobic and lyophilic particles: in some cases, even in media with high ionic strength (compressed EDL) their sols/suspensions remain dispersion stable because of the steric repulsion of the surface gel layers, with a thickness that grows with the surface charge density (pH-determined swelling moving away from the isoelectric point).

When choosing nanoparticles as anticancer drug carriers, both hydrophilicity/hydrophobicity and electric charges of adsorbents and adsorbates must be taken into account to ensure adsorption of the chemotherapeutic molecules and dispersion stability of the suspension before hydrogel formation. The solid-state particles considered in this review are separated into two classes: metal and dielectric (semiconductors have not been used so far) because of the presence or absence of free charges determines the adsorption ability of the nanoparticles; the dielectric particles are presented by oxides (metal-oxides, silicates, alumosilicates) and carbon (graphene and graphene oxide).

The particles are defined as nanoparticles when at least one dimension is on the nanometric scale or as a colloid when the size is submicrometric or micrometric; the distinction is made on the understanding that the surface atoms/molecules are different from those in the particle bulk because of uncompensated field-force, and this difference spans several atom/molecular layers. The particle size and shape determine the ratio of the surface (energetically different) to the bulk atoms/molecules; therefore, the needle- and disk-shaped particles with a nanometrically small size are defined as nanoparticles even when their big size is (sub)micrometric.

### 4.1. Metal Nanoparticles

The main difference of metal nanoparticles from dielectric ones is the free movement of electrons throughout the particle volume (electron gas). As a result, the electric field of the adsorbed molecules (created by their Coulomb and partial charges) does not penetrate in depth, but it is compensated in a thin layer of the metal surface (skin effect) in which the concentration of electrons is increased or decreased depending on the sign of the charges of the adsorbed molecules. The close distance between the charges of the metal and the molecules leads to a strong electrostatic attraction causing strong adsorption. This effect is enhanced by the intramolecular electron polarizability of anticancer chemotherapeutics, with molecules that are most often composed of benzene and heterocyclic compounds with conjugated double bonds (Section 2), with π-electrons that are delocalized and can move within the molecule.

The second feature of metal nanoparticles is that some atoms from their surface can be released into the medium in the form of positively charged ions, while the electrons remaining in the volume determine the negative charge of the particle. When a metal is placed in an aqueous medium, the direction of ions is always from the metal to the solution, but once equilibrium is reached, it is compensated by an opposing flow; the exchange current (the number of ions leaving or embedding in the crystal lattice per unit time per unit surface area) is different for different metals. The electric potential of a metal nanoparticle is determined by the concentration of its cations in solution (at high concentrations of salts of the same metal, the nanoparticle can become positively charged). As a result, an electrical double layer (EDL) is created at the metal/solution interface: the charge of the particle is compensated by ions of opposite sign (cations in excess of electrons in the metal) of the same type (potential determining) or another (indifferent, predominantly Na^+^ in biological tissues), which plays an important role in the adsorption of chemotherapeutics having a single or multiple elementary (Coulomb) electric charges; the surface electric potential of the particle facilitates or hinders adsorption due to electrostatic attraction or repulsion depending on the net electric charge (determined by the pH and the nature of the ionized groups) of the molecules. The surface charge density of the particles can be calculated from the ζ-potential (close to the surface potential), which is proportional to the electrophoretic mobility (velocity of the migration of charged particles in a direct electric field with a given strength) measured by the method of microelectrophoresis.

The third feature of metal nanoparticles with a pure (non-oxidized) surface is that it is hydrophobic due to its inability to form hydrogen bonds with water molecules (unlike metal oxides that have a hydrophilic surface). This creates favorable opportunities for the adsorption of hydrophobic chemotherapeutics, but this also requires avoiding aggregation of the nanoparticles. This can be achieved by the chemical modification of the surface (e.g., by oxidation or grafting of short-chain hydrophilic polymers), but most often by the adsorption of polymers already dissolved in the medium during the synthesis of the metal nanoparticles (in the case of classical gold and silver sol, a gelatin solution is used), or before their separation from the metal matrix is used for this purpose [102,103]. Suitable polymers are those with hydrophilic and hydrophobic groups (contacting the metal surface and the aqueous medium, respectively), which are adsorbed practically irreversibly thanks to multiple contacts secured by the flexibility of the polymer chain. The polymer layer provides water solubility and prevents aggregation of the nanoparticles due to steric repulsion (inability of the chains to interpenetrate each other). On the surface of the nanoparticles, free areas remain available for the adsorption of the relatively small molecules of the chemotherapeutic, since the surface is covered with polymer patches [104], the degree of occupation depends on the length and flexibility of the polymer chain (when it is stiffer, the free areas are larger). This means that the choice of polymer to stabilize metal nanoparticle suspensions should be based on the chemical nature and the way the units are linked (chemical grafting or physical adsorption), the hydrophilicity and flexibility of the chain (an example is the drastic difference between the rigid-chain cellulose and the flexible-chain amylose, two polymers composed of identical glucose units); molecular mass (chain length) is the second factor determining the steric repulsion between particles.

The chemical nature of nanoparticles and the equilibrium constant between the ion fluxes to/from the metal surface determine the physicochemical properties of nanoparticles, the most important of which are corrosion resistance and electrical charge. The exchange ion current at the silver/water interface is very high, and this determines their negative charge in water or physiological solution (0.15 M NaCl), whereas it is immeasurably low for the gold surface. By this reason, in aqueous medium, the gold nanoparticles are electrically neutral if they are prepared in oxygen-free (vacuum or inert gas) atmosphere; however, they acquire a small pH-dependent surface charge when they are synthesized in the presence of oxygen (atmospheric or dissolved in aqueous medium). Especially significant is the negative charge when the gold nanoparticles are synthesized in a water solution of oxygen-containing anions; then, the ζ-potential can reach some decade of millivolts (at neutral pH) and the sol of small gold nanoparticles can be kinetically stable due to the interparticle electrostatic repulsion, without need of additional stabilizers, as polymers angst aggregation. The emergence of a surface electric charge on gold nanoparticles is caused by the coordination of the oxygen donor atoms by back-bonding on the metal surface [105,106]. In particular, colloidal gold nanoparticles isolated by electrolyte-induced precipitation in citrate and tannic acid buffers have a negative surface charge due to the binding of the oxyanions: CO_3_^2−^, H_2_PO_4_^1−^, SO_4_^2−^ [107]. The ζ-potential diminishes with the concentration of NaCl because the indifferent Cl^−^ anions compete with the oxyanions; this leads to the aggregation of the gold seeds because of the reduced electrostatic repulsion (caused by both decreased surface electric potential and diminished EDL thickness) and allows for tuning the final size of the gold nanoparticles [108]. It can be noted that the ζ-potential of gold nanoparticles usually reported by the authors is not the original potential of the gold surface, but it reflects rather the charge of the adsorbed polyelectrolytes; when the adsorbed polymer is electrically neutral, the measured ζ-potential is diminished because of the additional hydrodynamic friction (caused by the polymer chains) when the particles migrate in the applied direct electric field.

The use of gold nanoparticles is driven by the corrosion resistance of gold (Au^0^) on the one hand, but on the other hand, the strong affinity of Au^0^ for sulfur (S) offers the possibility of chemical surface modification by coupling with sulfur-containing compounds as water-soluble alkanethiolates HS(CH_2_)*_n_*R (*n* ≥ 10), where the nature of group R determines the properties of the chemically modified surface: hydrophobic or hydrophilic with a positive or negative charge) [109,110]. The gold nanoparticles and hybrid core/shell particles (covered with a gold layer) [111] are most often used for biomedical applications [112,113,114,115,116,117], including as carriers of anticancer chemotherapeutics [118], an example are the hybrid Au–Fe-nanoparticles (magnetic Fe_3_O_4_-core and Au-shell) [119,120,121]).

For the above reasons, in composite hydrogels (composed from polymers, as shown in Table 2) those most commonly used as carriers for anticancer chemotherapeutics (Table 1) are nanoparticles from gold ore and silver (Table 3).

### 4.2. Oxide Nanoparticles

Solid-state oxide nanoparticles used in hydrogels as carriers of anticancer chemotherapeutics are chemical compounds of oxygen with metals, silicon and carbon. The surface properties of the particles are determined by the property of oxygen to bind with oxygen and hydrogen atoms from the aqueous environment forming hydrated boundary layers; the binding is by chemisorption, occurring mainly as specific chemical reactions with the formation of valent bonds with an energy of 40–400 kJ/mol and, therefore, is irreversible. Water molecules can only be removed by heating at a high temperature in anhydrous atmosphere. As a result of the chemisorption, a strong two-dimensional film of a chemical compound is formed on the particle surface, which does not penetrate deep into its solid phase. Examples are the surface oxides of carbon and graphite: (≡C)_2_O, (=C=O)_2_, (–C=O)_2_O, where the four-valent carbon atoms are covalently bound with the divalent oxygen atoms. In an aqueous medium, the first layer of water molecules is adsorbed irreversibly by chemical adsorption and the second by physical adsorption; the last is low-energy (4–40 kJ/mol) and therefore reversible, for example on quartz (silica): (–Si–O–Si–)O (dehydrated) → (–SiOH–O–SiOH–) (chemically adsorbed water) ↔ (–SiOH…OH_2_–O–SiOH…OH_2_–) (physically adsorbed water H_2_O), where the every four-valent Si atom is covalently bound with two O atoms and the hydroxyl groups SiOH form hydrogen bonds H…OH_2_ with the water molecules in the second layer of the hydrated boundary layer. Analogously, two hydration layers are formed on the surface of metal oxide (MO) nanoparticles in aqueous media; in the first layer, chemisorbed water molecules are irreversibly bonded via covalent bonds with the metal atoms on the solid surface (MOH), and the second is formed via reversible physical adsorption with hydrogen bonds. The majority of nanoparticles are initially hydrated, as they are synthesized in aqueous media by hydrolysis and polycondensation, while those synthesized in atmosphere are dehydrated, e.g., aerosil (Al_2_O_3_) particles synthesized by flame pyrolysis.

Thus, the presence of oxygen atoms in the structure of oxide nanoparticles provides chemical adsorption of the first layer of water molecules and the hydrophilicity of the surface by physical adsorption of water molecules from the medium. An example of this transition is the carbon structures: the surfaces of diamond, carbon, carbon black, graphite and graphene are hydrophobic owing to the inability of carbon atoms to form hydrogen bonds, but oxigraphene is hydrophilic due to the presence of oxygen atoms. Hydrophilicity is a determining factor in the selection of carrier particles for water-soluble anticancer chemotherapeutics. The second criterion is the electrical charge on the surface: it must be opposite for that of the adsorbed organic molecules for them to adsorb electrostatically. Since oxides are dielectrics (there are no free charges in the volume of the particles), the contribution of electrostatic forces is not as great as for adsorption on a metal surface. Surface roughness matters because the electric charges of oxide nanoparticles are not evenly distributed but are located on fixed centers. As a consequence, the chemical adsorption occurs predominantly on protruding bumps (where the solid-phase force field is stronger) and the physical adsorption is stronger in depressions (grooves, pores) where more of the surface atoms from different directions attract the adsorbed molecules.

The electrical properties of oxide nanoparticles (metal, silicate, carbonate) in aqueous media are determined by the ionization of hydroxyl groups or H^+^ proton joining, in particular, the surface of metal oxides is negatively charged:-MOH ↔ MO^−^ + H^+^, or positively: -MOH + H^+^ ↔ MOH_2_^+^, depending on the molar concentration of hydroxonium ions H_3_O^+^ in the aqueous medium (pH = −log [H_3_O^+^]) [135,136]. Analogously, charges on the surface of quartz ≡SiOH are produced by the ionization or adsorption of H^+^ cations. On the uncharged hydroxyl groups of oxides in aqueous media, metal-hydroxy complexes can be adsorbed: of the type M^z+^(OH)_z−1_^+^ (cations) or M^z+^(OH)_z+1_^−^ (anions), which charge the surface positively or negatively, respectively. In this way, the H^+^, OH^−^, M^z+^(OH)_z−1_^+^ and M^z+^(OH)_z+1_^−^ ions are potential-determining for the surface of oxide nanoparticles, and the ions with the highest concentration are indifferent in biological tissues: Na^+^, K^+^ and Cl^−^.

Fe_3_O_4_ iron oxide (magnetite) particles have ferromagnetic properties (they acquire a magnetic moment in an external magnetic field, unlike Fe_2_O_3_ particles that are superparamagnetic). The ferromagnetic properties do not affect the surface electrical properties of the particles, but their ability to magnetize makes them very convenient for manipulation in the preparation of composite particulate carriers of anticancer chemotherapeutics, in in vitro and in vivo experiments, and for manipulation in patients. 

The isoelectric point (or point of zero charge) of particles with a pH-dependent surface charge is observed at pH, either when all surface groups are uncharged (when there is only one kind of surface center) or when the two kinds of charges are equal (1/2 positive and 1/2 negative), as are amphoteric surfaces, in particular those of oxides. In the absence of the specific adsorption of indifferent ions, the isoelectric point coincides with the point of zero charge, otherwise they diverge to a lower or higher pH depending on the type (cation or anion) of ions adsorbed specifically (involving chemical forces in addition to electrical ones). In Table 4, the isoelectric points of oxide nanoparticles used as carriers of anticancer chemotherapeutics are given.

In Table 5, Table 6 and Table 7, composite hydrogels with included metal oxide, silica and graphene nanoparticles are given; in the case of hybrid particles, the covering substances (the shell) are mentioned because they determine their adsorption ability.

### 4.3. Choice of Nanoparticles as Carriers of Anticancer Chemotherapeutics

The adsorption of chemotherapeutics onto oxide nanoparticles occurs via hydrogen bonding, electrostatic and van-del-Waals forces. Therefore, the choice of particles should be based primarily on their hydrophilicity/hydrophobicity and the presence of ionizable groups with integer (Coulomb) charges in the chemotherapeutic molecule. For insoluble or poorly water-soluble organic molecules, metal (especially gold) nanoparticles are suitable, whereas for water-soluble ones, oxide nanoparticles (metal oxides, silicates, carbonates) are suitable. In the case that water-soluble chemotherapeutics have ionizable groups, the pH should be chosen so that the surface centers of the particles (charged positively or negatively) and the chargeable groups of the organic molecules have opposite electric charges to condition the electrostatic adsorption on the surface. The contribution of electrostatic forces in the case of the ion-induced dipole type is usually negligible for small molecules, but may be significant for chemotherapeutics with organic molecules that are composed of bonded benzene nuclei because the adjoined double bonds allow for the migration of π-electrons over significant distances (the induced dipole moment is equal to the charge × distance).

## 5. Cytotoxicity of Nanoparticles

Separately, nanoparticles (without any chemotherapeutics) are demonstrated to have a toxic effect not only on cancer cells, but also on normal ones. There are various mechanisms involved in the cytotoxicity of the nanoparticles.

### 5.1. Direct Mechanical Interaction

The flat particles with a nanometric thickness (such as graphene oxide and montmorillonite), when performing a Brownian motion can hit the cell membrane and disrupt its integrity by their sharp edges, leading to a decrease in the transmembrane ion gradient, and by that indirectly blocking the action of Na^+^-K^+^-ATPase and ATP synthesis, respectively; this has been demonstrated on bacterial colonies of *Escherichia coli* [208]. This effect may explain the antibacterial effect of montmorillonite (bentonite, widely used for human weight loss), the oral administration of which leads to the disruption of the normal intestinal flora (dysbacteriosis) [209].

### 5.2. Free Radicals Generation and Oxidative Stress

Oxide nanoparticles (some metal oxide nanoparticles such as copper oxide, zinc oxide and others, as well as graphene oxide nanoparticles) can generate reactive oxygen species (ROS) such as superoxide anion-radical O_2_^–•^, perhydroxyl radical HO_2_^–•^, hydrogen peroxide H_2_O_2_, hydroxyl-radical HO^•^ and singlet oxygen ^1^O_2_ [210,211]. These free radicals cause lipid peroxidation of the cell membrane phospholipids, damage to the DNA [212], depolymerization of the polysaccharides as well as protein oxidation. Under physiological conditions, there are two types of antioxidant systems: (a) non-enzymatic: ascorbic acid (vitamin C), α-tocopherol (vitamin E), β-carotenoids, glutathione, etc.; and (b) enzymatic: superoxide dismutase (SOD), catalase (CAT), glutathione peroxidase (GSHPx) that neutralize and inactivate free radicals. When the balance between the free radicals formation and their neutralization by biological antioxidant systems is disturbed, oxidative stress occurs, leading to severe cell damage [213,214].

Copper oxide nanoparticles have been shown to have the highest cytotoxic effect in comparison with zinc oxide, ferric oxide and titanium dioxide [215,216]. It is also demonstrated that graphene oxide nanoparticles can generate ROS induced by visible light [217] and cause oxidative stress, which suggests their antibacterial activity [218]. Graphene oxide nanoparticles have a genotoxic effect: DNA fragmentations and chromosomal aberrations are observed [219].

### 5.3. Disruption of the Cell Communication

Graphene oxide nanosheets (hydrophilic) and aggregates of graphene (hydrophobic), when partially covering the cell membrane (being adsorbed on it), can hinder the transport processes to/from the cell, and isolate the cell biologically from the environment. This puts the cell in a trap, as it cannot acquire the substances necessary for its functioning, as well as excrete the unnecessary products from its metabolism [220].

### 5.4. Heavy Metal Ions Release

Some metal oxide nanoparticles release heavy metal ions that have a cytotoxic effect because of enzyme inhibition; the effect is caused by electrostatic and/or coordinated binding of the strongly charged metal cations (with a valency that is from two to six, for example: copper Cu^2+^, gadolinium Ga^3+^, chromium Cr^2+^, Cr^4+^, Cr^6+^), with the negatively charged reactive center of some enzyme macromolecules, or destruction of their 3D structure (denaturation). For example, zinc oxide/graphene oxide (ZnO/GO) hybrid nanoparticles have been demonstrated to release zinc cations (Zn^2+^), which have an antibacterial effect on *E. coli* culture, while the cytotoxic effect on the HeLa cell line was not significant; the authors suppose that it is because of the difference in the Zn^2+^-dependent prokaryotic and eukaryotic deaths [221].

### 5.5. In Situ Generation of Oxygen (O_2_) Nanobubbles

It has been recently shown that reduced graphene oxide/zinc peroxide-silver (rGO)/ZnO_2_-Ag) nanoframeworks with pH- and temperature-depending behavior can generate oxygen (O_2_) nanobubbles and, respectively, hydrogen peroxide (H_2_O_2_) through the Fenton-like pathway [222,223]; as a result, this nanoframework has antibacterial activity against methicillin-resistant *Staphylococcus aureus* (MRSA), *Staphylococcus aureus* and *Escherichia coli*.

## 6. Conclusions

Because of their biocompatibility, suitable physicochemical properties and biodegradability (via solubilization and chemical or enzymatic hydrolysis) of the majority of them, hydrogels are widely used as a drug delivery system. Since their drug absorption capacity is restricted due to the poor adsorption of hydrophilic and particularly hydrophobic molecules by the gel polymers, different nanoparticles (metal, metal oxide, silicates, graphene oxide, etc.) can be integrated into the hydrogels in order to enhance their absorption capacity. These nanoparticles, owing to their high dispersibility conditioned by their small size, have a huge specific surface area and consequently a huge adsorption capacity, which enable them to adsorb and carry various chemotherapeutics (such as doxorubicin, paclitaxel, etc.). Non-covalent bonds (hydrogen bonds, electrostatic and van-der-Waals forces) are crucial for the adsorption of chemotherapeutics onto nanoparticles. Consequently, the selection of particles should be primarily based on their hydrophilicity/hydrophobicity and the presence of ionizable groups in the chemotherapeutic molecule. Metal (mainly gold) nanoparticles are appropriate for insoluble or poorly water-soluble organic molecules, while oxide nanoparticles (metal oxides, silicates, carbonates) are suitable for water-soluble ones. If water-soluble chemotherapeutics have ionizable groups, the pH of the medium should be selected to ensure that the surface of the nanoparticles and the groups of the organic molecules have opposite charges to interact electrostatically.

In order to more effectively create composite hydrogels with higher cytotoxicity, greater consideration must be given to the physicochemical mechanisms and particle parameters discussed in this review, which have not been sufficiently addressed by the authors when composing the particles, and therefore have often not achieved a satisfactorily high cytotoxicity. As cytotoxicity depends on a number of parameters: type of hydrogel, chemotherapeutic, type of particle (size, shape, charge, hydrophilicity/hydrophobicity), in vitro or in vivo experiment, type of cell line, it is necessary to conduct studies where only one of these parameters varies in order to be able to draw fully relevant conclusions about which type of hydrogel, chemotherapeutic and particle is most suitable for the treatment of a given type of cancer.

## Figures and Tables

**Figure 1 gels-09-00421-f001:**
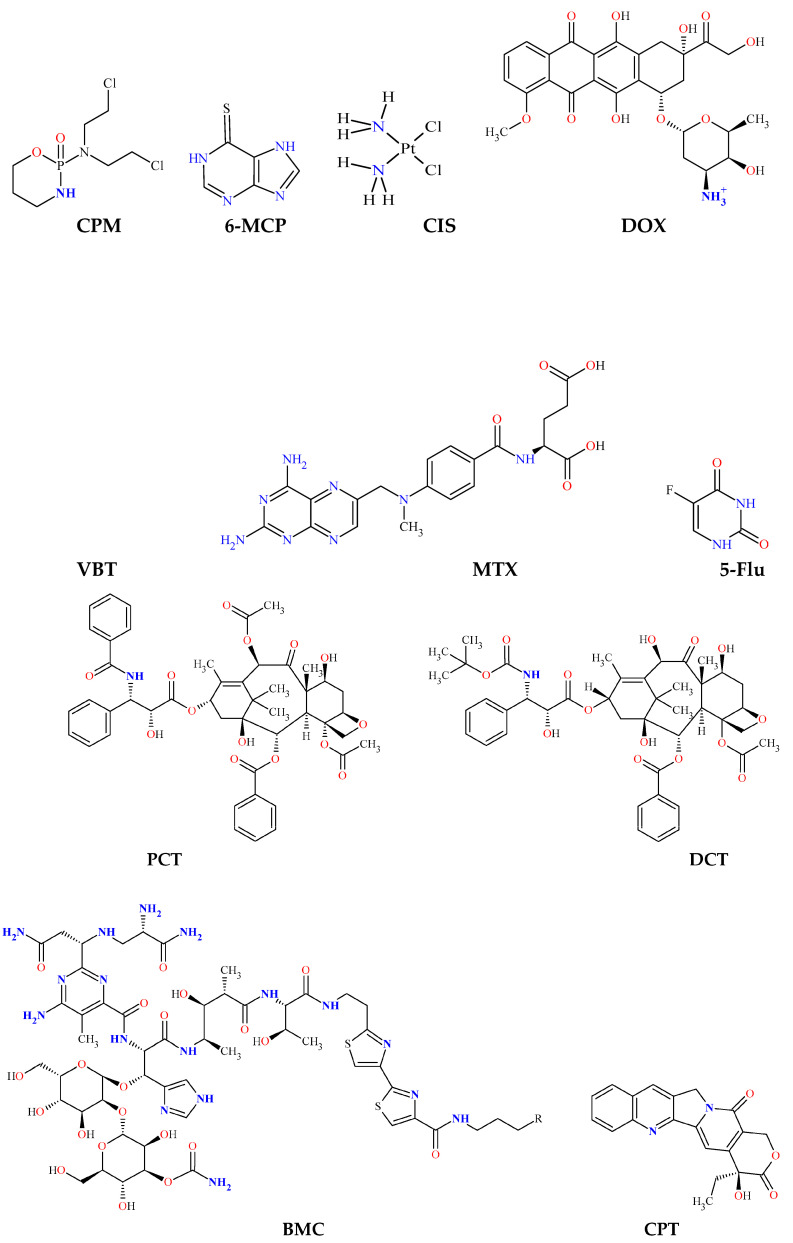
Chemical structures and abbreviations of some chemotherapeutics used as anticancer drugs by composition of hydrogel with included nanoparticles.

**Figure 2 gels-09-00421-f002:**
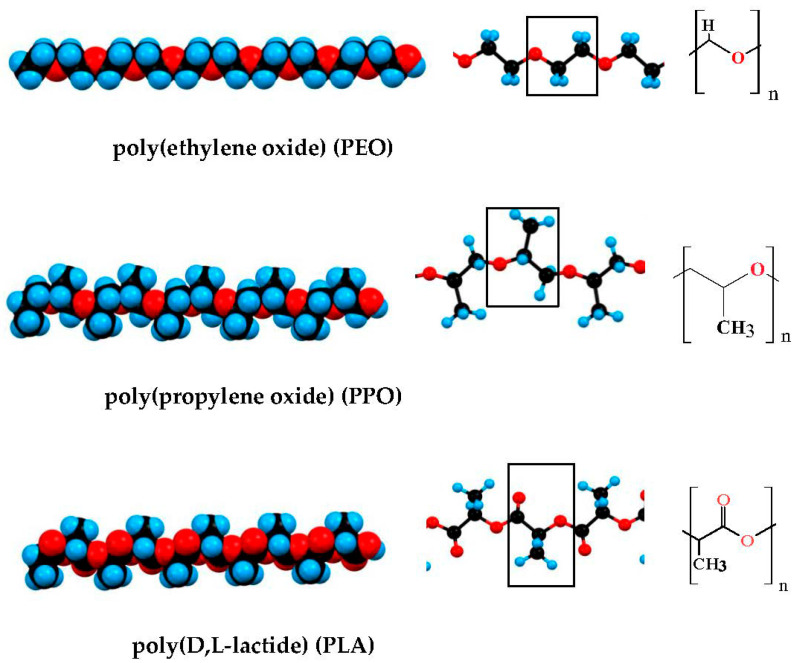
Polymers most commonly used for composition hydrogels with included nanoparticles carrying different chemotherapeutics. * dendrimer.

**Table 1 gels-09-00421-t001:** pK_a_ and solubility of some anticancer chemotherapeutics.

Chemotherapeutic	Abbreviation	pK_a_		Solubility in Water, 25 °C	Refs.
Methotrexate	MTX	4.7		1 mg/mL	[37,38,39]
Camptothecin	CPT	4.7		0.0027 mg/mL	[40,41]
Cisplatin	CIS	5.4	7.2	1 mg/mL	[42,43]
Vinblastine	VBT	5.4	7.4	10 mg/mL	[44,45,46]
5-Fluorouracil	5-Flu	8.0		12.2 mg/mL	[47,48,49]
Bleomycin	BMC	7.3	7.5	20 mg/mL	[50,51]
Doxorubicin	DOX	8.9	9.9	50 mg/mL	[52,53,54]
Paclitaxel	PCT	10.0		≤0.0001 mg/mL	[55,56,57,58]
Docetaxel	DCT	10.7		0.0019 mg/mL	[59,60]
6-Mercaptopurine	6-MCP	11.2		0.734 mg/mL	[61,62,63]
Cyclophosphamide	CPM	12.1		40 mg/mL	[64,65]

**Table 2 gels-09-00421-t002:** Polymers used for hydrogels. The abbreviations repeat those used by the authors of the cited references.

№	Polymer	Abbreviation
1	gelatin	G
2	hyaluronic acid	HA
3	alginate	ALG
4	chitosan	CS
5	dextran	DEX
6	oleopolyol	OA
7	poly(N-isopropylacrylamide)	PNIPAm
8	Poly (N, N-diethyl acrylamide)	PDEA
9	polyacrylonitrile-polyamide	PAN-PA
10	poly(acrylic acid)	PAAc
11	poly(amidoamine)	PAAm
12	poly(methacrylic acid)	PMAA
13	poly(N-isopropylacrylamide-co-acrylamide	PNIPAAm
14	poly(N-vinylpyrrolidone)	PVPON
15	poly(*N*-isopropylacrylamide-*co*-acrylamide)	PNIPAAm-*co*-AAm
16	poly(β-aminoester urethane)	PAEU
17	acrylamide-methylenebisacrylamide-green tea	AM-MBA-GT
18	poly(polypropylene glycol)	PPG
19	poly(ethylene glycol)	PEG
20	methoxypoly(ethylene glycol)	mPEG
21	polyethyleneimine	PEI
22	carboxymethyl cellulose	CMC
23	poly lactic-co-glycolic acid	PLGA
24	poly(ethylene glycol) -oleic acid	OA-PEG
25	prepare aminated guar gum	AGG
26	polyethyleneimine	PEI
27	poly vinyl alcohol	PVA
28	poly(N-isopropylacrylamide)	PNIPAM
29	poly(β-amino ester)	PBAE
30	poly(N-isopropylacrylamide-co-maleic anhydride)]@strach	PNIPAAm-co-MA @starch
31	poly(ethylene glycol)-block-poly(N-isopropylacrylamide- co-maleic anhydride)_2_-graft-poly(ethylene glycol)	PEG-b-(PNIPAAm- co-PMA)_2_-g-PEG
32	*N*,*N*′-(dimethylamino)ethyl methacrylate-co-maleic anhydride	DMAEMA-*co*-MA
33	poly(*N*-isopropylacrylamide-*co*-itaconic anhydride)-	P(NIPAAm-*co*-IA)-PEG
34	poly[(2-succinyloxyethylmethacrylate)-*b*-(*N*-isopropyl acrylamide)-*b-*dimethylaminoethylmethacrylate)	P(SEMA-*b-*NIPAM- *b*-DMAEMA)
35	glycidylmethacrylate-grafted-maleated cyclodextrin	P(GMA-g-MACD)
36	poly(D,L-lactide-co-glycolide)-b-poly(ethylene glycol)-b-poly(D,L-lactide-co-glycolide)	PLGA-PEG-PLGA
37	sodium alginate- poly(acrylamide- co-N-vinylcaprolactam-co-acrylamidoglycolic acid)	SA-PAVA
38	poly (N-vinyl pyrrolidone/dextran)	PVP-DEX
39	*Strychnos potatorum* L. (SPL) polysaccharide-based dual-responsive semi-IPN-type	SPL-DMA
40	N-fluorenylmethoxycarbonyl-di-phenylalanine	Fmoc-FF
41	poly(N-isopropylacrylamide-co-acrylamide)	NIPAAm-co-AAm
42	poly(*N*-isopropyl-acrylamide-acrylic acid)	PNA

**Table 3 gels-09-00421-t003:** Hydrogels with silver and gold nanoparticles for the delivery of anticancer chemotherapeutics. The abbreviations of the drugs and the gel-forming polymers are given in Table 1 and Table 2, respectively.

N	Polymers	Nanoparticles		Drug	Loading	Releasing	Cytotoxicity	Ref.
		Metal	Size					
1	CMC	silver	sphere 10 nm	DOX	noncovalent	pH	60%	[122]
2	SA-PAVA	silver	sphere 20 nm	5-Flu	noncovalent	pH	–	[123]
3	PVP-DEX	silver	sphere 12 nm	DOX	noncovalent	pH	–	[124]
4	SPL-DMA	silver	sphere 20 nm	DOX 5-Flu	noncovalent	pH	85%	[125]
5	PAAc	gold	sphere 5 nm	DOX	noncovalent	pH	–	[126]
6	PEG	gold	sphere 13 nm	DOX	noncovalent	pH	in vivo	[127]
7	Fmoc-FF	gold	sphere 226 nm	DOX 5-Flu	noncovalent	T	–	[128]
8	NIPAAm-co-AAm	gold	sphere 150 nm	DOX	noncovalent	NIR	30%	[129]
9	PNA	gold	cubic 50 nm	DOX	noncovalent	NIR	75%	[130]
10	PEG-CS	gold	rod 54 nm	PCT	noncovalent	NIR	in vivo	[131]
11	ALG	gold	–	CIS	noncovalent	T	in vivo	[132]
12	ALG	gold	sphere 100 nm	CIS	noncovalent	–	66%	[133]
13	PNIPAAm	gold	sphere 50 nm	5-Flu	noncovalent	Ph, T	70%	[134]

**Table 4 gels-09-00421-t004:** Isoelectric point (IEP) of some oxide nanoparticles.

	Oxide-Nanoparticles	IEP	Ref.
1	SiO_2_	2	[137]
2	Mesoporous silica	2–3	[138]
3	Cr_2_O_3_	3	[139]
4	SnO_2_	3.8	[140]
5	Fe_3_O_4_	5.0	[141]
6	γ-Fe_2_O_3_	5.5	[142]
7	TiO_2_	6.4	[143]
8	CuO	6.5	[144]
9	α-Fe_2_O_3_	6.7	[139]
10	ZrO_2_	7.0	[145]
11	CeO_2_	8.0	[146]
12	γ-Al_2_O_3_	8.5	[147]
13	α-Al_2_O_3_	9.2	[148]
14	Mn_2_O_3_	9.8	[149]
15	NiO	10	[150]
16	ZnO	10.3	[151]
17	MgO	12–12.7	[152]

**Table 5 gels-09-00421-t005:** Hydrogels are composed of metal oxide nanoparticles for the delivery of chemotherapeutics: magnetic (Fe_3_O_4_, consisting also of iron(II) oxide FeO and iron(III) oxide Fe_2_O_3_), zinc oxide (ZnO), copper oxide (CuO), manganese oxide (MnO) nanoparticles. The abbreviations of the drugs are given in Table 1, and those of the (co)polymers are given in Table 2 (repeating the abbreviations used by the authors in the corresponding references); *T*—temperature.

N	Polymers	Nanoparticles		Drug	Loading	Releasing	Cytotoxicity	Ref.
		Metal	Size					
1	G, ALG	Fe_3_O_4_	sphere 25 nm	DOX	noncovalent	pH	60%	[153]
2	PNIPAAm-co-MA@starch	Fe_3_O_4_	sphere 70 nm	DOX	covalent	pH	–	[154]
3	PEG-b-(PNIPAAm-co-PMA)2-g-PEG	Fe_3_O_4_	sphere 100 nm	DOX	noncovalent	pH	–	[155]
4	DMAEMA-*co*-MA	Fe_3_O_4_	sphere 130 nm	MET	covalent	pH	65%	[156]
5	P(NIPAAm-*co*-IA)-PEG	Fe_3_O_4_	sphere 168 nm	DOX	noncovalent	pH	90%	[157]
6	P(SEMA-*b-*NIPAM-*b*-DMAEMA)	Fe_3_O_4_	sphere 22 nm	DOX	noncovalent	pH	80%	[158]
7	PEI, CMC	Fe_3_O_4_	sphere 15 nm	DOX	noncovalent	pH	–	[159]
8	poly(γ-GA-co-γ-GAOSu)-g-PEG-FA, γ-PGA	Fe_3_O_4_	sphere 43 nm	DOX	noncovalent	pH	70%	[160]
9	OA-PEG	Fe_3_O_4_	sphere 234 nm	DOX	noncovalent	–	50%	[161]
10	ALG, G	Fe_3_O_4_	sphere 113 nm	DOX	noncovalent	magnetic	80%	[162]
11	Agar	Fe_3_O_4_	sphere 10 nm	DOX	–	*T*	85%	[163]
12	ALG	Fe_3_O_4_	sphere 700 nm	DOX	–	pH	90%	[164]
13	AGG	Fe_3_O_4_	sphere 16 nm	DOX	noncovalent	–	–	[165]
14	PVPON	Fe_3_O_4_	sphere 20 nm	DOX	noncovalent	pH	–	[166]
15	PAAc	Fe_3_O_4_	sphere 70 nm	DOX	noncovalent	pH	–	[167]
16	G, PVA PLGA	Fe_3_O_4_	sphere 2–10 μm	DOX	noncovalent	–	55%	[168]
17	Chitosan	Fe_3_O_4_	sphere	DOX	noncovalent	pH	–	[169]
18	DEX	Fe_3_O_4_	sphere 20 nm	DOX	noncovalent	pH	–	[170]
19	PBAE	Fe_3_O_4_	sphere 20 nm	PTX	noncovalent	*T*	55%	[171]
20	mPEG	Fe_3_O_4_	sphere 20 nm	PTX	noncovalent	*T*	–	[172]
21	ALG	Fe_3_O_4_	sphere	5-Flu	noncovalent	pH	–	[173]
22	CS PAAc	Fe_3_O_4_	sphere 98 nm	5-Flu	noncovalent	pH	–	[174]
23	mPEG–LUT	Fe_3_O_4_	sphere	5-Flu	noncovalent	pH *T*	–	[175]
24	AM-MBA-GT	Fe_3_O_4_	sphere 10 nm	5-Flu	noncovalent	pH *T*	–	[176]
25	P(GMA-g-MACD)	Fe_3_O_4_	sphere 20 nm	5-Flu	noncovalent	pH	55%	[177]
26	CMC	ZnO	sphere 20 nm	DOX	noncovalent	*T*	60%	[178]
27	PVA-oxidized starch	CuO	sphere 45 nm	DOX	noncovalent	*T*	70%	[179]
28	Cellulose-PAA	MgO	rod	DOX	noncovalent	pH	70%	[180]
29	PMAA PVPON	mMnO	sphere cubic 2.4 μm	DOX	noncovalent	pH	40%	[181]

**Table 6 gels-09-00421-t006:** Hydrogels are composed of silica nanoparticles for the delivery of anticancer chemotherapeutics. The abbreviations of the drugs and the (co)polymers are given in Table 1 and Table 2, repeating those used by the authors in the corresponding references; *T*—temperature; NIR—near infrared radiation.

N	Polymers	Nanoparticles		Drug	Loading	Releasing	Cytotoxicity	Ref.
		Type	Size					
1	PEG PPG	mSiO_2_	sphere 60 nm	DOX	noncovalent	dissolution	in vivo	[182]
2	HA	mSiO_2_	sphere 45 nm	DOX	covalent	enzymatic	90%	[183]
3	Peptide	mSiO_2_	sphere 90 nm	DOX	noncovalent	*T*	–	[184]
4	PNIPAAm- co-AA	Au-SiO_2_	sphere 326 nm	DOX	noncovalent	NIR	90%	[185]
5	PMAA	SiO_2_	sphere	DOX	noncovalent	pH	80%	[186]
6	PMAA	SiO_2_	sphere 100 nm	DOX	noncovalent	pH	–	[187]
7	HA	mSiO_2_	sphere 200 nm	DOX	noncovalent	pH enzymatic	80%	[188]
8	PAN-PA	SiO_2_	sphere 20 nm	DOX	noncovalent	pH	–	[189]
9	HA-azobenzene	mSiO_2_	sphere 150 nm	DOX	noncovalent	*T*	90%	[190]
10	ALG	mSiO_2_	sphere 130 nm	DOX	covalent	–	90%	[191]
11	PEG, PPG, HA	SiO_2_	sphere 198 nm	CIS	noncovalent	*T*	71%	[192]
12	CS, PVA, OA	SiO_2_	sphere 50 nm	CIS	noncovalent	pH	65%	[193]
13	PEG-PAEU	mSiO_2_	sphere 157 nm	CPT	noncovalent	pH	80%	[194]
14	CS, CMC, HA	NH_2_-mSiO_2_	sphere 300 nm	CYT MTX	covalent	pH	90%	[195]
15	PNIPAm	LAM	plate	5-Flu	noncovalent	*T*	–	[196]
16	CS	MM	plate 270 nm	DOX	noncovalent	pH	88%	[197]
17	CS	MM	plate 140 nm	DOX	noncovalent	pH	66%	[198]
18	ALG	MM	plate 142 nm	DOX	noncovalent	pH	75%	[199]
19	PLGA-PEG-PLGA	MM	plate	DOX	noncovalent	*T*	in vivo	[200]

**Table 7 gels-09-00421-t007:** Hydrogel are composed of graphene oxide or graphene nanoparticles for the delivery of anticancer chemotherapeutics. The abbreviations of the drugs are given in Table 1, and those of the (co)polymers are given in Table 2 (repeating the abbreviations used by the authors in the corresponding references); NIR—near infrared radiation.

N	Polymers	Nanoparticles		Drug	Loading	Releasing	Cytotoxicity	Ref.
		Type	Size					
1	CS, PMAA	graphene oxide	–	DOX	noncovalent	pH	75%	[201]
2	PEI	graphene oxide	sphere 320 nm	DOX	noncovalent	pH	in vivo	[202]
3	Acrylated-CS NIPAM PEG-diacrylate	graphene oxide	sphere 320 nm	DOX	noncovalent	NIR	75%	[203]
4	CS	graphene oxide	sphere 20 nm	DOX	noncovalent	pH	40%	[204]
5	CS-cellulose	graphene oxide	–	DOX	noncovalent	pH	in vivo	[205]
6	PDEA	graphene	sphere 15 nm	DOX	noncovalent	pH	in vivo	[206]
7	– (aerogel)	graphene oxide	sheet 200 nm	DOX PTX	noncovalent	pH	–	[207]

## Data Availability

Not applicable.

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
