# Peer review of "Composite Hydrogels with Included Solid-State Nanoparticles Bearing Anticancer Chemotherapeutics"

_gels, 2023, doi:10.3390/gels9050421_

Round 1

Reviewer 1 Report

A review regarding chemotherapy of cancer using hydrogel nanomaterials has been presented. The subject is interesting and valuable. However, there are some points which should be considered by the authors for further completion of the review, as commented below:  

1.       Table 7 can be further completed and supported by “Graphene aerogel nanoparticles for in-situ loading/pH sensitive releasing anticancer drugs”.

2.       One of the important agents in applications of nanoparticles in cancer therapy is personalized protein corona effect. See, for example, [Biomater. Sci., 2017, 5, 378-387] and [Nanoscale 7 (2015) 8978-8994]. This subject should be addressed and discussed in the revised version.

3.       Nanoparticles can induce some toxic effects during the cancer therapy (both on the normal and cancerious cells). The mechanisms involved in the cytotoxicity should be mentioned and discussed in the revised version. For further help to authors, the well-known mechanisms involved in the cytotoxicity of nanomaterials are: 1) physical direct interaction of extremely sharp edges of nanomaterials with cell wall membrane [Toxicity of graphene and graphene oxide nanowalls against bacteria], 2) ROS generation [RSC Adv., 2015,5, 80192-80195] even in dark [Langmuir 2015, 31, 33, 9155–9162], 3) trapping the bacteria within the aggregated nanomaterials [Wrapping bacteria by graphene nanosheets for isolation from environment, reactivation by sonication, and inactivation by near-infrared irradiation], 4) oxidative stress [ACS Nano 2011, 5, 9, 6971–6980], 5) interruption in the glycolysis process of the cells [Escherichia coli bacteria reduce graphene oxide to bactericidal graphene in a self-limiting manner], 6) DNA damaging [Free Radical Biology and Medicine Volume 51, Issue 10, 15 November 2011, Pages 1872-1881] and geno-toxicity [Size-dependent genotoxicity of graphene nanoplatelets in human stem cells], 7) heavy metal ion release [ACS Appl. Mater. Interfaces 2014, 6, 4, 2791–2798], and recently 8) contribution in generation/explosion of nanobubbles [Oxygen-Rich Graphene/ZnO2-Ag nanoframeworks with pH-Switchable Catalase/Peroxidase activity as O2 Nanobubble-Self generator for bacterial inactivation].

4.       The manuscript needs outlook and conclusion section.

5.       Could the authors compare the influence of the importing nanomaterials in the efficiency of the anticancer chemotherapeutics? This can be done by designing a table.

6.       The authors should discuss about the magnetic nanoparticles applied in cancer therapy via drug delivery. See, for example, [Drug Metabolism Reviews, Volume 52, 2020 - Issue 1, Pages 205-224].

7.       The authors should discuss on the biodegradability of the hydrogels used in drug delivery.

It is acceptable. 

Author Response

AZ: Thank you very much for your competent and highly professional review, which helped us exceptionally to improve the quality of our article. All amendments and additions in the revised manuscript are given in blue.

A review regarding chemotherapy of cancer using hydrogel nanomaterials has been presented. The subject is interesting and valuable. However, there are some points which should be considered by the authors for further completion of the review, as commented below: 

  1. Table 7 can be further completed and supported by “Graphene aerogel nanoparticles for in-situ loading/pH sensitive releasing anticancer drugs”.

AZ: In Table 7 the suggested article about graphene aerogel nanoparticles is added (N7).

  1. One of the important agents in applications of nanoparticles in cancer therapy is personalized protein corona effect. See, for example, [Biomater. Sci., 2017, 5, 378-387] and [Nanoscale 7 (2015) 8978-8994]. This subject should be addressed and discussed in the revised version.

AZ: In the revised version of the introduction the personalized protein corona effect is addressed and discussed.

  1. Nanoparticles can induce some toxic effects during the cancer therapy (both on the normal and cancerious cells). The mechanisms involved in the cytotoxicity should be mentioned and discussed in the revised version. For further help to authors, the well-known mechanisms involved in the cytotoxicity of nanomaterials are: 1) physical direct interaction of extremely sharp edges of nanomaterials with cell wall membrane [Toxicity of graphene and graphene oxide nanowalls against bacteria], 2) ROS generation [RSC Adv., 2015,5, 80192-80195] even in dark [Langmuir 2015, 31, 33, 9155–9162], 3) trapping the bacteria within the aggregated nanomaterials [Wrapping bacteria by graphene nanosheets for isolation from environment, reactivation by sonication, and inactivation by near-infrared irradiation], 4) oxidative stress [ACS Nano 2011, 5, 9, 6971–6980], 5) interruption in the glycolysis process of the cells [Escherichia coli bacteria reduce graphene oxide to bactericidal graphene in a self-limiting manner], 6) DNA damaging [Free Radical Biology and Medicine Volume 51, Issue 10, 15 November 2011, Pages 1872-1881] and geno-toxicity [Size-dependent genotoxicity of graphene nanoplatelets in human stem cells], 7) heavy metal ion release [ACS Appl. Mater. Interfaces 2014, 6, 4, 2791–2798], and recently 8) contribution in generation/explosion of nanobubbles [Oxygen-Rich Graphene/ZnO2-Ag nanoframeworks with pH-Switchable Catalase/Peroxidase activity as O2 Nanobubble-Self generator for bacterial inactivation].

AZ: A new section (Section 5. Cytotoxicity of nanoparticles) regarding the mechanism of the cytotoxicity of the nanoparticles has been included in the revised version of the manuscript.

  1. The manuscript needs outlook and conclusion section.

AZ: The conclusion is added in the revised version of the manuscript.

  1. Could the authors compare the influence of the importing nanomaterials in the efficiency of the anticancer chemotherapeutics? This can be done by designing a table.

AZ: As the cytotoxicity depends on a number of parameters: type of hydrogel, chemotherapeutic, type of particle (size, shape, charge, hydrophilicity/hydrophobicity), in vitro or in vivo experiment, type of cell line, it is impossible to make a relevant comparison with the currently available date in the literature.

  1. The authors should discuss about the magnetic nanoparticles applied in cancer therapy via drug delivery. See, for example, [Drug Metabolism Reviews, Volume 52, 2020 - Issue 1, Pages 205-224].

AZ: Magnetic (Fe3O4) are discussed in Table 5 and in Section 4.2.

  1. The authors should discuss on the biodegradability of the hydrogels used in drug delivery.

AZ: A new section (Section 3.3. Biodegradability of hydrogels) is included in the revised version of the manuscript.

Reviewer 2 Report

1. Line 43-44, one typical study (Journal of Materials Chemistry B 5 (5), 935-943) should be included to support hydrogels as carriers for chemotherapeutic agents.

2. The figure captions are too simplified.

3. Line 286, the authors should discuss why Ca2+ or Zn2+ would adsorb on the PEO chains.

4. One figure containing studies related to hydrogels loaded with chemotherapeutic drugs should be added.

5. A section of conclusion and perspective should be added.

6. Technical issue. Line 31, '[1, 2]'; Line 247 and 290, '[]'. Please check all.

Minor editing of English language required

Author Response

AZ: Thank you very much for the professional review, which helped us exceptionally to improve the quality of our article. All amendments and additions in the revised manuscript are given in blue.

  1. Line 43-44, one typical study (Journal of Materials Chemistry B 5 (5), 935-943) should be included to support hydrogels as carriers for chemotherapeutic agents.

AZ: The reference is added in the revised version of the manuscript.

  1. The figure captions are too simplified.

AZ: The figures captions are supplemented in the new version of the manuscript.

  1. Line 286, the authors should discuss why Ca2+ or Zn2+ would adsorb on the PEO chains.

AZ: This statement is supported by a study (European Polymer J. 34, (1998), 531-538)

  1. One figure containing studies related to hydrogels loaded with chemotherapeutic drugs should be added.

AZ: Our review is focused on composite hydrogels with included solid-state nanoparticles carrying chemotherapeutics; pure hydrogels with chemotherapeutic drugs (without included nanoparticles) and composite hydrogel with soft-meta particles are not the subject of the present review article.

  1. A section of conclusion and perspective should be added.

AZ: The conclusion is included in the revised version of the manuscript.

  1. Technical issue. Line 31, '[1, 2]'; Line 247 and 290, '[]'. Please check all.

AZ: All technical issues throughout the manuscript are corrected.

Reviewer 3 Report

The manuscript is well documented, and contains a lot of useful information, mainly for graduate or PhD students commencing their research in the field. I have a single important observation:

1. I miss illustrations (not just descriptions) of results of significance, selected among the different combinations described. Mostly, such results should belong to the authors' own work, justifying their writing of a review in the field.

I also point out some minor, formal issues:

a) Title: I would change solid-state by simply "solid"

b) Lines 260 261: XXXX??

c) Line 333, line 340, line 529, line 527: Ref. [ ] ?

d) All throughout the paper: the electrical double layer is traditionally abbreviated EDL not DEL

e) Lines 557ff: cyrillic characters

f) Table 4: mesopores -> mesoporous

g) No Conclusions required by the journal?

Author Response

AZ: Thank you very much for the professional review, which helped us exceptionally to improve the quality of our article. All amendments and additions in the revised manuscript are given in blue.

The manuscript is well documented, and contains a lot of useful information, mainly for graduate or PhD students commencing their research in the field. I have a single important observation:

  1. I miss illustrations (not just descriptions) of results of significance, selected among the different combinations described. Mostly, such results should belong to the authors' own work, justifying their writing of a review in the field.

AZ: We are not specialist in the field of hydrogels, but we have years of experience with water suspensions of solid-state colloid particles. Therefore, our review article is focused on such nanoparticles emphasizing their physicochemical properties. Our review article was written in response to an invitation from the Editors.

I also point out some minor, formal issues:

  1. a) Title: I would change solid-state by simply "solid"

AZ: We use the term “solid-state” as it is generally accepted in the physics of the condense matter.

  1. b) Lines 260 261: XXXX??

AZ: All technical issues throughout the manuscript are corrected.

  1. c) Line 333, line 340, line 529, line 527: Ref. [ ] ?

AZ: All technical issues throughout the manuscript are corrected.

  1. d) All throughout the paper: the electrical double layer is traditionally abbreviated EDL not DEL

AZ: The abbreviation is corrected in the revised version of the manuscript.

  1. e) Lines 557ff: cyrillic characters

AZ: All technical issues throughout the manuscript are corrected.

  1. f) Table 4: mesopores -> mesoporous

AZ: The word in Table 4 is corrected.

  1. g) No Conclusions required by the journal?

AZ: The conclusion is added in the revised version of the manuscript.

Round 2

Reviewer 1 Report

The authors well addressed all of the comments in the revised version. Therefore, the manuscript is ready for publication. However, there is still a minor point which should be considered by the authors as mentioned below:

In the revised version this statement has been added: “Other polymers that undergo enzymatic hydrolysis are: chitosan, gelatin, etc [94]”. However, this needs to be improved and supported as follows: “Other polymers that undergo enzymatic and/or pH-sensitive hydrolysis/gelation are: chitosan, gelatin, etc [94] & [DOI: 10.4172/2157-7552.1000212].”.

After considering this point, the manuscript can be considered for publication in Gels.

It is acceptable. 

Author Response

Thank you very much for your second review.

In the revised version of the manuscript the suggested statment is improvenved with the mentioned reference (marked in yellow). 

Reviewer 3 Report

The authors have responded adequately to my previous comments. I recommend publication of this review

Author Response

Thank you very much for your recommendation.